# Tannic Acid Improves Renal Function Recovery after Renal Warm Ischemia–Reperfusion in a Rat Model

**DOI:** 10.3390/biom10030439

**Published:** 2020-03-12

**Authors:** Louise Alechinsky, Frederic Favreau, Petra Cechova, Sofiane Inal, Pierre-Antoine Faye, Cecile Ory, Raphaël Thuillier, Benoit Barrou, Patrick Trouillas, Jerome Guillard, Thierry Hauet

**Affiliations:** 1INSERM, U1082 IRTOMIT, 86021 Poitiers, France; l.alechinsky@gmail.com (L.A.); sofiane.inal@gmail.com (S.I.); cecile.ory@chu-poitiers.fr (C.O.); rathuillier@gmail.com (R.T.); benoit.barrou@gmail.com (B.B.); 2Université de Limoges, Faculté de Médecine, EA 6309 “Maintenance Myélinique et Neuropathies Périphériques”, 87025 Limoges, France; frederic.favreau@unilim.fr (F.F.); pierreantoine.faye@gmail.com (P.-A.F.); 3CHU de Limoges, Laboratoire de Biochimie et Génétique Moléculaire, 87042 Limoges, France; 4University Palacký of Olomouc, RCPTM, Dept Physical Chemistry, Faculty of Science, 771 46 Olomouc, Czech Republic; petra.cechova@upol.cz; 5CHU de Poitiers, Laboratoire de Biochimie, 86021 Poitiers, France; 6Université de Poitiers, Faculté de Médecine et de Pharmacie, 86073 Poitiers, France; 7Département Hospitalo-Universitaire de Transplantation SUPORT, 86021 Poitiers, France; 8Inserm, UMR 1248, Fac. Pharmacy, Univ. Limoges, 87025 Limoges, France; patrick.trouillas@unilim.fr; 9Université de Poitiers, UMR CNRS 7285 IC2MP, Team 5 Organic Chemistry, 86073 Poitiers, France; jerome.guillard@univ-poitiers.fr

**Keywords:** warm ischemia, cold ischemia, tannic acid, renal function recovery, oxidative stress

## Abstract

Background and purpose: Ischemia–reperfusion injury is encountered in numerous processes such as cardiovascular diseases or kidney transplantation; however, the latter involves cold ischemia, different from the warm ischemia found in vascular surgery by arterial clamping. The nature and the intensity of the processes induced by ischemia types are different, hence the therapeutic strategy should be adapted. Herein, we investigated the protective role of tannic acid, a natural polyphenol in a rat model reproducing both renal warm ischemia and kidney allotransplantation. The follow-up was done after 1 week. Experimental approach: To characterize the effect of tannic acid, an in vitro model of endothelial cells subjected to hypoxia–reoxygenation was used. Key results: Tannic acid statistically improved recovery after warm ischemia but not after cold ischemia. In kidneys biopsies, 3 h after warm ischemia–reperfusion, oxidative stress development was limited by tannic acid and the production of reactive oxygen species was inhibited, potentially through Nuclear Factor erythroid-2-Related factor 2 (NRF2) activation. In vitro, tannic acid and its derivatives limited cytotoxicity and the generation of reactive oxygen species. Molecular dynamics simulations showed that tannic acid efficiently interacts with biological membranes, allowing efficient lipid oxidation inhibition. Tannic acid also promoted endothelial cell migration and proliferation during hypoxia. Conclusions: Tannic acid was able to improve renal recovery after renal warm ischemia with an antioxidant effect putatively extended by the production of its derivatives in the body and promoted cell regeneration during hypoxia. This suggests that the mechanisms induced by warm and cold ischemia are different and require specific therapeutic strategies.

## 1. Introduction

Ischemia is the cessation of blood flow to an organ and reperfusion is its restoration. The combination of both processes induces a range of lesions at the cellular and organ level, leading to increased cell death and immunogenicity, assembled under the term Ischemia–Reperfusion Injury (IRI). This pathology is widely encountered in medicine, particularly during vessel and heart surgery and is unavoidable in organ transplantation. Indeed, all organs commonly require transport from the donor to the recipient, cold flushing with a preservation solution, and maintenance of hypothermia (approx. 4 °C) during transport (from 2 to 36 h). IRI induces several pathways, such as inflammation, cellular death by necrosis or apoptosis and oxidative stress, particularly during the reperfusion stage [1,2]. Oxidative stress is an imbalance between reactive oxygen species (ROS) production and elimination by antioxidant systems.

Tannic acid is a plant-derived polyphenol from the family of hydrolysable tannins. It is composed of a penta-*O*-galloyl-*beta*-*D*-glucose (PGG) nucleus and several gallic acid moieties. It has shown several beneficial effects, such as reducing serum cholesterol and triglyceride levels, as well as antioxidant activities [3,4,5]. The multiple phenol groups at its periphery contribute to the stabilization of its oxidized (radical) forms, via electronic delocalization in the π-conjugated system. This makes tannic acid and its derivatives efficient ROS scavengers, in turn reducing IRI [4,6] (Figure 1).

The cellular antioxidant defense system is composed of many enzymes. In particular, Nuclear Factor erythroid-2-Related factor 2 (NRF2) is a transcription factor sensitive to ROS and a key player in oxidative metabolism [7]. Interestingly, this transcription factor can be induced by some polyphenols [8].

In the present study, we hypothesized that tannic acid, metabolized to gallic acid and PGG, will provide lasting protection against IRI. We also hypothesized that such protection could differ between warm and cold ischemia.

## 2. Materials and Methods

### 2.1. Determination of Blood Tannic Acid and Derivatives Levels after Intraperitoneal Administration

To determine the bioavailability of tannic acid after injection and to identify the kinetics of its metabolite formation, gallic acid concentrations were determined in rat blood. After intraperitoneal injection of tannic acid (50 mg/kg), blood was collected in the tail vein at 0.5, 1, 2, 4 and 24 h. Gallic acid determination was performed by HPLC (Hitachi LaChrom^®^, column C18X Terra MS, Waters) associated to spectrophotometry, according to the method previously described with some modifications [9,10]. Briefly, 0.25 mL of plasma were spiked with 0.05 mL of internal standard (2,3-dimethoxybenzoic acid, 10 mg/mL), and 1 mL of 0.1 M acetate buffer (pH = 5) (Sigma aldrich), 0.05 mL of a 4% EDTA-Na_2_ solution (Merck), 0.1 mL of 0.6 M CaCl_2_ solution, and 500 UI of β-glucuronidase (G0751-1MU, Sigma aldrich). The samples were incubated for 45 min at 37 °C. Enzymatic hydrolysis was stopped by the addition of 0.5 mL of 1N sulfuric acid and component were extracted twice with 4 mL of ethyl acetate. After evaporation, the residue was dissolved in 0.2 mL of ethyl acetate and evaporated again and dissolved in 0.2 mL of mobile phase (acetonitrile/NaH_2_PO_4_ buffer pH 3.6; 30/70; *v*/*v*). The chromatogram analysis conditions are flow rate 0.7 mL/min (isocratic), run time: 10 min.

### 2.2. Animal Care and Use

Animal experimental procedures were performed in accordance with the French Ministries of Agriculture and Research guidelines and approved by the institutional committee for the use and care of laboratory animals (CEEA N°84 Poitou-Charentes, project reference number: 2015112414598711 and CEEA N°34, project reference number: 15048).

#### 2.2.1. Renal Warm Ischemia–Reperfusion Procedure

Sixty-four rats were randomized into eight groups of eight animals. The 7 d groups designate animals with 7 d of reperfusion after ischemia (sacrificed on day seven after IR) and the 3 h groups designate animals with 3 h of reperfusion (sacrificed 3 h after IR). The IR groups designate rats which underwent bilateral renal ischemia–reperfusion and injection of a vehicle solution according to the protocol described below. The groups IR + tan designate the rats which underwent bilateral renal ischemia–reperfusion and injection of a tannic acid solution. The groups Sham 3 h, Sham 7 d, Sham + tan 3 h, Sham + tan 7 d designate rats with the same surgery, but without a renal ischemia–reperfusion sequence with the injection of vehicle solution or tannic acid (Appendix A).

Male Sprague–Dawley rats (Janvier Labs, Saint Berthevin, France), from 265 to 350 g, were used. They were maintained in a climate-controlled room at 22 ± 2 °C and a humidity of 60 ± 5%, on a light/dark cycle, and provided a standard commercial diet and water ad libitum. After an adaptation period of three days, rats were anesthetized by isoflurane inhalation (induction 4% and maintenance 1.5%) associated with ketamine intraperitoneal injection (12.5 mg/kg). Rats were then randomly injected intraperitoneally with tannic acid (50 mg/kg) or the vehicle (NaCl 0.9%). Bilateral renal ischemia was performed 30 min after injection, by right and left renal pedicles clamping using appropriate vascular clamps (85 g pressure, Fine Science Tools, Micro Serrefins FST#18055-02, Heidelberg, Germany). After 60 min, clamps were removed. Good vascularization of the kidneys was visually checked by the coloration change. Blood collection was performed by the tail vein. At the time of euthanasia, the kidneys were collected and biopsies performed for tissue analysis (Appendix A).

#### 2.2.2. Renal Cold Ischemia in Transplantation Procedure

Thirty-five rats were randomized into four groups:

-KT (Kidney Transplantation) (*n* = 12): kidney transplantation with a bilateral nephrectomy and vehicle solution injection;

-KT + tan (*n* = 11): kidney transplantation with a bilateral nephrectomy and tannic acid solution injection;

-Sham (*n* = 6), and Sham + tan. (*n* = 6): rats with a dissection of the left renal pedicle and ureter and right nephrectomy with vehicle solution or tannic acid.

Male Lewis rats (Janvier Labs, Saint Berthevin, France), from 250 to 350 g, were used with similar housing conditions as above. After an adaptation period of three days, rats were anesthetized by isoflurane inhalation (induction 4% and maintenance 1.5%). In the donor rat, after dissection of the left renal artery and vein, the ureter was sectioned and the kidney dissected. Then, ligation of the infrarenal aorta and the suprarenal aorta were performed and the renal vein was ligated and cut. An arteriotomy was performed and a 25-gauge needle was inserted into the aorta. The kidney was flushed with 10 mL of SCOT 15 at 4 °C and the renal artery was sectioned. The kidney graft was placed in 30 mL of SCOT 15 at 4 °C for 6 h. The recipient rats were then randomly injected intraperitoneally with tannic acid (50 mg/kg) or vehicle solution (NaCl 0.9%). Thirty minutes after injection, kidney transplantation began. The surgical time, including bilateral nephrectomy and anastomosis, did not exceed 60 min. After dissection of the left renal artery and vein in the recipient, the ureter was sectioned and the kidney dissected. Then, clamping of the renal vessels was performed. The artery and vein were cut as close as possible to the hilum. The renal graft was set in place of the recipient left kidney and end-to-end vascular anastomoses were performed. The donor ureter was anastomosed with the ureter of the recipient with an end-to-end anastomosis. Right nephrectomy was then performed. Sham procedure consisted in a dissection of the renal pedicle and the ureter with a mobilisation of the kidney and a right nephrectomy. Blood collection was performed by the tail vein. At the time of euthanasia, the kidneys were collected and biopsies performed for tissue analysis (Appendix A).

### 2.3. Plasma Creatinine Determination

Plasma creatinine concentrations were measured on the Cobas C701 automatic analyzer (Roche Diagnostics, Meylan, France) by the compensated Jaffé technique.

### 2.4. Glutathione and Superoxide Dismutase Assays

Total glutathione (GSH) and superoxide dismutase (SOD) activity within kidney biopsies were measured, respectively, with the glutathione assay kit (Bertin Pharma, Orléans, France) and the SOD activity kit (Bertin Pharma), following the manufacturer’s instructions. Protein levels in homogenate were determined by the Bicinchoninic acid protein assay kit (Sigma Aldrich, St. Quentin Fallavier, France).

### 2.5. Immunohistochemistry and Western Blot Analysis

Briefly, frozen sections on renal cortex samples were used for Cell ROX assay and NRF-2 staining (Ref: Ab31163, 1/100, Abcam, Paris, France) detected by immunofluorescence. Protein levels were determined through western blotting according to standard protocols [11], using samples from frozen cortical kidney biopsies and antibodies detecting Xanthine Oxidase (Xanthine oxidase, Ref: SC-20991, 1/400, Santa Cruz Biotechnology, Santa Cruz, United States), P67-Phox (Nicotinamide Adenine Dinucleotide Phosphate Oxidase (NADPH) subunit, Ref: 07-002, 1/375, Millipore, Molsheim, France), NRF-2 (Ref: Ab31163, 1/1000, Abcam) and loading control: GlycerAldehyde-3-Phosphate DeHydrogenase (GAPDH) (1/3000; Millipore).

### 2.6. Cytokine Levels on Blood

Interleukin 6 (IL-6), Interleukin 10 (IL-10) and tumor necrosis factor alpha (TNFa) levels were determined in blood with Enzyme-Linked ImmunoSorbent Assay (ELISA) kit assays (Quantikine, RD Sytem, Minneapolis, United States) following the manufacturer’s instructions.

### 2.7. Effects of Tannic Acid and Derivatives on ROS Production in Human Aortic Endothelial Cells Subjected to Hypoxia–Reoxygenation

To mimic the ischemia–reperfusion found in vivo, we used a standard protocol of hypoxia–reoxygenation in human aortic endothelial cells (HAEC, Gibco, Courtaboeuf, France), cultured as recommended by the manufacturer [12]. Cells were synchronized, then treated with increasing concentrations of tannic acid, gallic acid, PGG or Phosphate Buffer Saline (PBS) during hypoxia (6 h) and reoxygenation (4 or 18 h) at 37 °C (Appendix A). Cell death was determined at the end of experiment using the MultiTox-Fluor Multiplex Cytotoxicity Assay (Promega™, Charbonnieres, France) and ROS quantification by the Cell ROX assay. The fluorescence of Cell ROX assay was determined by image J software on seven fields obtained from an Olympus BX41 microscope (Rungis, France) and reported to the number of cells evaluated by DAPI fluorescence. Cell migration was also investigated by scratch assay.

### 2.8. Theoretical Methodology

To obtain the necessary parametrisation for the molecular dynamics (MD) simulations, tannic acid was split into two parts, the PGG (central) moiety and the gallic acid (phenolic) tails, both with an acetyl cap. Both moieties were optimised with the B3LYP/6-31G* method with the Gaussian software. The optimised structures were used for charge calculation with the HF/6-31G* method. The partial charges were obtained using the Amber RESP procedure performed according to the official Amber manual. The structures of the sugar moiety and the gallic acid tail were then aligned in a pairwise manner in the PyMol software, using the carbonyl carbon, carbonyl oxygen and ether oxygen. The caps were removed, and the structures were connected between sugar carbonyl carbon and the corresponding aromatic ring carbon of the tail. The gaff2 forcefield parameters were used for the tail part of the molecule and the GLYCAM forcefield was used for the sugar moiety. To check the parametrisation and stability of the molecule, a 50 ns run was performed in water. The same starting structure of the molecule was then inserted into a simulation box containing an equilibrated solvated 1-palmitoyl-2-oleoylphosphatidylcholine (POPC) bilayer solvated in water, and the whole system was minimised. The simulations were conducted with three different replicas, for 1µs each. Gallic acid was modelled in the POPC bilayer in a similar way to tannic acid. Gallic acid was inserted in the POPC membrane box and the whole system was simulated for 500 ns, which allowed sufficient sampling of insertion events and diffusion of gallic acid in the lipid bilayer. To evaluate the impact of the protonation state, gallic acid was simulated in its two major protonation states, namely the fully protonated one (the neutral form denoted COOH) and the deprotonated one (the anion form denoted COO^-^). The deprotonation was at the carboxyl group, as the pKa of the COOH group of gallic acid is around 4.3, while the phenolic OH groups have pKa ranging from nine to 13 [13,14].

### 2.9. Statistical Analysis

We used NCSS 2007 (Hintze, J. 2007, NCSS, LLC. Kaysville, UTAH). The values shown are means ± SD. We used Newman–Keuls or Kruskal–Wallis tests for multiple comparisons or a Mann–Whitney test for comparisons between two groups. Statistical significance was considered for *p* < 0.05.

## 3. Results

### 3.1. Plasma Kinetics of Tannic Acid Derivative after Intraperitoneal Injection

A high level of gallic acid (around 19 mg/L) was detected in the plasma 1h after injection and remained detectable at 24 h (Figure 2), indicating that the maximum concentration was reached 1 h after injection. Thus, this time window was used to test the effectiveness of tannic acid against renal IRI.

### 3.2. Protective Effects of Tannic Acid on Renal Function Recovery after Warm Renal Ischemia–Reperfusion

Intraperitoneal injection of tannic acid 30 min before surgery improved renal function recovery after 60 min of renal ischemia: at 24 h post ischemia, plasma creatinine levels in the group without tannic acid injection increased up to 300 µmol/L, while treated animals showed a mild increase (approximately 100 µmol/L). After 7 d, the IR + tan group showed a full recovery of renal function compared to the untreated group (Figure 3).

### 3.3. Ineffectiveness of Tannic Acid on Renal Function Recovery after Renal Transplantation

According to the literature, 6 h of cold ischemia was chosen [15] to induce sufficient injuries but minimize death. Tannic acid injection was ineffective at improving kidney graft function recovery, as similar levels of creatinine were obtained in each group (Figure 4).

### 3.4. Tannic Acid Injection Reduces Oxidative Stress Induced by Warm Ischemia–Reperfusion

Kidney biopsies at 3 h after warm ischemia-reperfusion were collected and analyzed. Tannic acid limited the expressions of major pro-oxidant enzymes such as xanthine oxidase and p67 phox, a subunit of NADPH oxidase involved in superoxide anion production (Figure 5A,B). In addition, tannic acid limited the decrease in glutathion levels post-ischemia–reperfusion, without any effect on SOD activity (Figure 5C,D). Cell ROX staining showed that ROS production induced by warm ischemia–reperfusion was significantly abolished by tannic acid (Figure 6). Immunostaining revealed enhanced NFR2 expression by tannic acid, mainly in the nucleus (Figure 7A,B). However, the total expression of NRF2 in the kidneys was not different between groups (Figure 7C).

### 3.5. Tannic Acid and Derivatives Protect Endothelial Cells from Hypoxia–Reoxygenation Injury

The mechanism of tannic acid protection was characterized using a cellular model of endothelial cell subjected to warm hypoxia–reoxygenation. Tannic acid was added to the culture medium during hypoxia and reoxygenation steps, mimicking the rat treatment. A dose-dependent reduction of cytotoxicity was observed with the different tannic acid derivatives (Figure 8A). These results were supported by XTT results obtained under the same conditions (Figure 8B). They were also supported by the reduction of ROS production by tannic acid and its main derivative, gallic acid, at the concentration of 7.3 µmol/L (Figure 9). The cell migration assay suggested that tannic acid improved endothelial cell migration during warm hypoxia. This difference was not found after 24 h of reoxygenation (Figure 10).

### 3.6. Positioning of Tannic Acid in Membranes

The capacity of tannic acid to be inserted into membranes, and then to act as an antioxidant to protect against lipid oxidation, was evaluated by MD simulations performed on in silico models of membranes solvated in water. The model was made of POPC lipids, being one of the most representative phospholipids present in the organism. Tannic acid was primarily positioned in the bulk water, where MD simulations revealed a large structural variability. Several stable tannic acid conformations were identified, mainly driven by π–π stacking interactions (Figure 11). Within the first 200 ns of the MD simulations, tannic acid was inserted in the membrane. It reached a relatively stabilized position, which was, however, conformation dependent. By averaging over all MD simulations, the clustering analysis identified six major conformations of tannic acid in the membrane, of which all six spent over 90% of the simulation time (Figure 12). For one conformer, the PGG moiety was inserted deeper than the five gallic acid tails, in close contact with the glycerol moiety and the upper part of the lipid tails (Replica 3, Figure 13). Apart from this conformer, such a deep insertion of the PGG moiety was prevented sterically by the gallic acid tails. The PGG was anchored in the region of the phosphates, whereas the gallic acid tails were distributed either in close contact with water and the choline moieties from one side, or in close contact with the glycerol moiety and the upper part of the lipid tails from the other side (Figure 13).

Both the protonated and deprotonated forms of gallic acid interact with the membrane, albeit in slightly different manners. The protonated (neutral) form moves quickly at the surface of the membrane and is positioned in close contact with the glycerol moiety and the upper part of the lipid tails; during the simulation, it occasionally reaches the middle of the lipid tails. The deprotonated (anionic) form quickly reaches the surface of the membrane; however, it is positioned further from the membrane centre, fluctuating from the water phase to the glycerol moiety, with only little contact with the lipid tails (Figure 14).

## 4. Discussion

The potential of plant-derived medicine in the fields of aging, cancer, neurodegenerative disorders and cardiovascular diseases has attracted increasing interest in past decades. However, the beneficial effect is linked to an appropriate concentration in the body in the long term. Tannic acid is a polyphenol found frequently in plants. It is known for its pharmacologic properties, such as antiseptic, veinotonic, bactericides and antioxidant [16,17,18], the latter being clearly related to its polyphenolic chemical features [19]. Studies showed that the antioxidant activity of tannic acid could have beneficial effects in vivo on diseases like iron-overload hepatotoxicity, by scavenging in vivo ROS and chelating iron [20]. In addition, tannic acid inhibits the generation of ROS in irradiation-induced apoptosis in a megakaryocytes model, preventing apoptosis, promoting platelet recovery and raising survival [21]. In the body, tannic acid is metabolized into several derivatives, such as gallic acid or PGG, already known for their beneficial effects against oxidative stress and IRI [22,23].

In this study, we used a rat model of renal warm ischemia by renal pedicle clamping [24]. We first determined the level of tannic acid in the rat blood after an intraperitoneal injection of 50 mg/kg, suggesting an effective concentration of tannic acid around 12.5 mg/L (7.3 µmol/L) to protect endothelial cells from hypoxia–reoxygenation. After injection, tannic acid was not found in the blood, but its main derivative, gallic acid, was increased after 30 min with a peak at 1h and a residual concentration at 24 h. This suggested that 50 mg/kg intraperitoneally 30 min before surgery is adapted to obtain an effective concentration 1 h 30 min after injection, at the time of kidney reperfusion. We showed that tannic acid is also metabolized into other derivatives (potentially PGG), unidentified with our technique but with a peak concentration at around 24 h, suggesting a prolonged effect of tannic acid (data not shown).

After renal warm ischemia–reperfusion, animals were followed for 1 week. Renal recovery was significantly improved by tannic acid: the increase in plasma creatinine at day one was reduced and returned to basal levels at day seven. Plasma creatinine was not affected by tannic acid in the sham group, supporting a lack of nephrotoxicity.

In order to investigate the efficiency of tannic acid in transplantation, a rat model of allo-transplantation was used. However, the injection of tannic acid was ineffective in improving renal recovery. These results could be explained by the intensity of injury induced by transplantation, higher than that of warm ischemia, even with conditions adapted to promote animal survival at day seven. Allo response could be also involved in this context with the lack of immunosuppressive therapy. In this regard, Lewis are known as particularly syngeneic and are able to minimize the effects of immune rejection related to transplantation. On the other hand, in warm ischemic protocol, we choose Sprague–Dawley rats because of their sensitivity to renal warm ischemia, allowing 7 d survival [15,24,25,26,27,28]. Taken together, these results suggest that the protective mechanism of tannic acid could be efficient in warm ischemia and limited during cold ischemia–reperfusion.

In order to investigate the mechanism of action of tannic acid regarding antioxidant activity, we explored the redox status 3 h after reperfusion. This time was chosen because oxidative stress is currently described at the beginning of reperfusion although some studies have described it during ischemia [24]. Interestingly, tannic acid injection reduced the expression of major enzymes involved in ROS production. It limited the expression of xanthine oxidase, an enzyme producing superoxide anions after its activation by intracellular calcium release [1,29] and p67 phox, a subunit of NADPH oxidase, found in activated leucocytes, producing superoxide anions. This suggests that tannic acid has a beneficial role in reducing the production of ROS over a long period of time and, in our case, during renal IRI. The concentration of glutathione, the main intracellular redox buffer, was increased with tannic acid treatment. This result supports a stimulation of antioxidant defense. Quantification of ROS production by Cell ROX staining in kidney biopsies confirms these results.

NRF2 is a transcription factor activated by ischemia–reperfusion but also by polyphenolic compounds [8]. After release to the complex formed with keap-1, NRF2 is translocated to the nucleus to stimulate Antioxidant Response Elements (ARE) sequences and stimulates the expression of antioxidant enzymes. In our conditions, protein NRF2 levels were not affected by warm ischemia–reperfusion with or without tannic acid treatment. However, at 3 h after reperfusion, we observed increased translocation of NRF2 to the nucleus of the renal cells, supporting a process in development.

In order to decipher if tannic acid or its derivatives were efficient against IRI, we used an in vitro model of endothelial cells subjected to hypoxia–reoxygenation. Endothelial cells are the first cells affected by ischemia–reperfusion injury and are also involved in inflammatory and regeneration processes [29,30]. As suggested by our in vivo results, tannic acid was efficient to significantly limit cytotoxicity. In addition, the major derivatives of tannic acid, gallic acid and PGG were also efficient in a dose-dependent manner and protected against ROS production. These results strongly suggest that tannic acid or its derivatives are able to scavenge ROS and limit cytotoxicity, and support our hypothesis that the duration of tannic acid effectiveness observed in vivo could be extended by gallic acid formation and other derivatives. Interestingly, tannic acid stimulated cell migration and proliferation during hypoxia, a step where oxidative stress is minor, supporting that, in addition and independently to its antioxidant action, tannic acid could act through another mechanism to stimulate survival.

As with many other polyphenols, tannic and gallic acids are efficient free radical scavengers due to the presence of the galloyl moiety, allowing efficient H-atom transfer to the free radicals [19]. However, to be efficient, antioxidants are required to insert into biological membranes, which are a major site of ROS production and propagation. To evaluate the capacity of both tannic and gallic acids to reach this site, we performed MD simulations, which provide an atomistic description of membrane insertion. Both compounds were prone to insertion into the membrane, but through different methods. Because of its amphiphilic character, tannic acid efficiently interacts with biological membranes, allowing lipid oxidation inhibition. Because of its bulky character and the presence of the polar PGG central moiety, the gallic acid moieties of tannic acid cannot reach deep regions of the membrane. Conversely, because of its small size and amphiphilic character, the neutral form of the derivative gallic acid can reach deeper regions of the lipid bilayers, where lipid oxidation and lipid peroxidation propagation are likely to occur. It must be stressed that, due to its low first pKa value, gallic acid exists in water in its anionic form. As for other compounds, it is likely to approach the membrane in this form and reprotonation occurs inside the membrane where the acid-base balance is likely shifted.

In conclusion, our results show that the antioxidant tannic acid exhibits protective effects on renal recovery after warm ischemia, as found during vascular surgery. This is particularly interesting considering the fact that tannic acid is a natural compound, easily accessible in nature, and is soluble in aqueous solution, which is rare for polyphenols. However, its beneficial role was not observed in kidney transplantation, potentially explained by the intensity of injury or by the additional alloimmune response. The beneficial activity after warm ischemia–reperfusion could be linked to the formation of active derivatives in the body, which offers an extended period of protection against ROS production up to 24 h, limiting cytotoxicity and promoting regeneration to accelerate renal recovery. In addition to the ROS scavenging activity of polyphenols, we suggested a stimulation of endothelial cell proliferation associated to NRF2 process induction. This study underlined that the protection by tannic acid has to be applied during ischemia and also during reperfusion, and that the mechanisms induced by warm and cold ischemia could be different and require adapted therapeutic strategies.

## Figures and Tables

**Figure 1 biomolecules-10-00439-f001:**
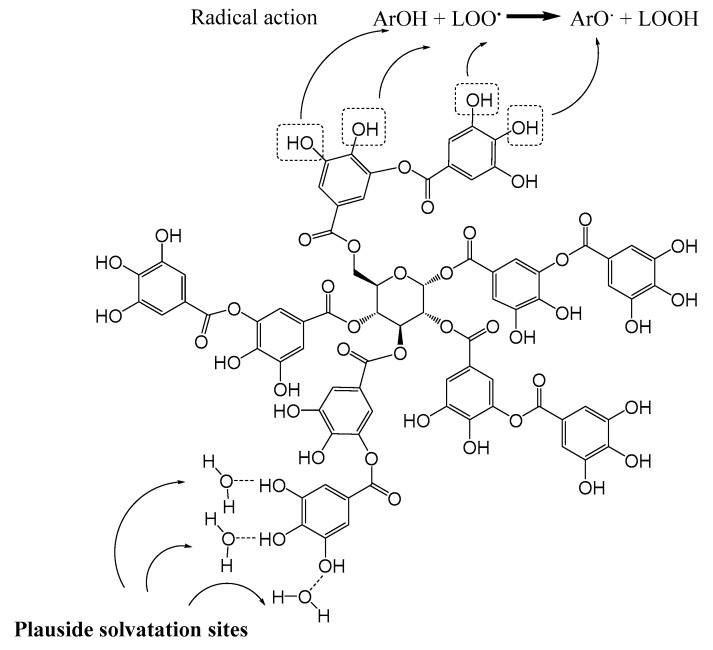
Chemical formula of tannic acid. Tannic acid has numerous phenol groups supporting a high solubility in aqueous solutions and a great antioxidative capacity. The phenolic groups (ArOH) give electrons to different radicals (LOO°) formed during ischemia–reperfusion and the radical tannic acid produced (ArO°) remains stable and not reactive linked to tautomeric forms produced from the benzenic group.

**Figure 2 biomolecules-10-00439-f002:**
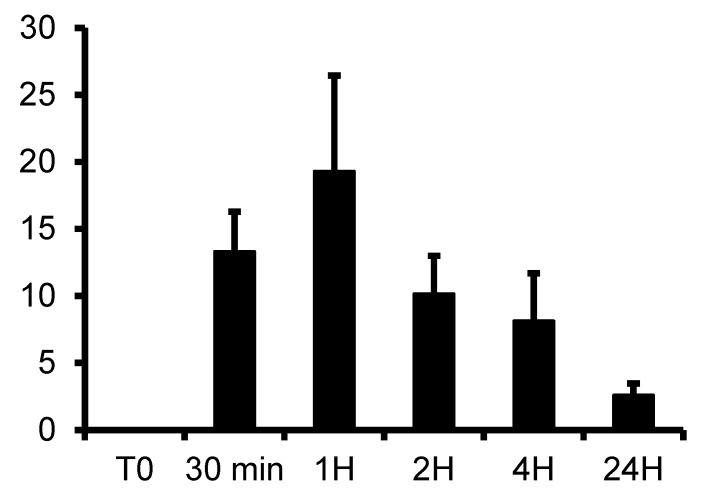
Plasma kinetic of gallic acid after intraperitoneal injection of tannic acid at 50 mg/kg in rat. Gallic acid was determined in plasma collected from 30 min to 24 h, by high pressure liquid chromatography method coupled with spectrophotometry, *n* = 3–4 at each times. Values are mean ± SD.

**Figure 3 biomolecules-10-00439-f003:**
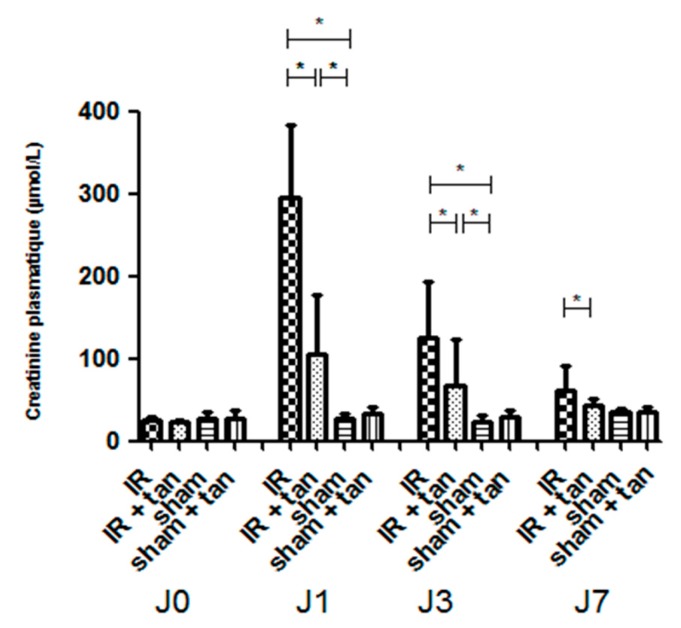
Tannic acid injection improves renal function recovery after warm kidney ischemia– reperfusion. Rats were subjected to sham or bilateral renal ischemia for 60 min by renal pedicle clamping with or without prior injection of tannic acid. Renal function was evaluated by creatinine plasma levels concentrations. *n* = 8, values are mean ± SD, * p < 0.05.

**Figure 4 biomolecules-10-00439-f004:**
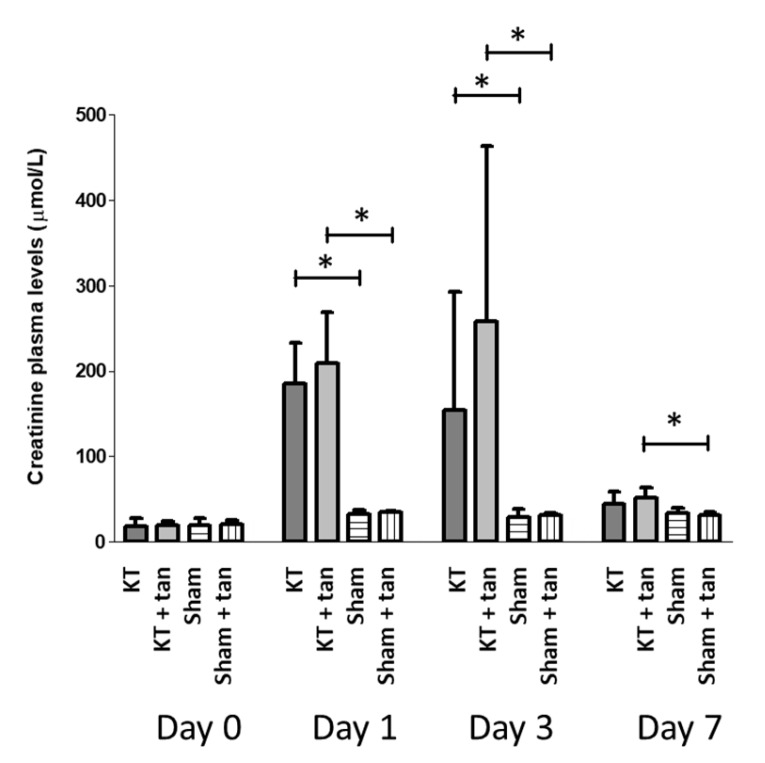
Tannic acid injection was ineffective to improve renal function recovery after cold ischemia– reperfusion in kidney graft model with 6 h of cold preservation. Rats were subjected to sham or kidney transplantation with bilateral nephrectomy with or without prior injection of tannic acid. Renal function was evaluated by creatinine plasma levels concentrations. *n* = 7–8 in transplanted groups and *n* = 6 in sham groups, values are mean ± SD, * p < 0.05.

**Figure 5 biomolecules-10-00439-f005:**
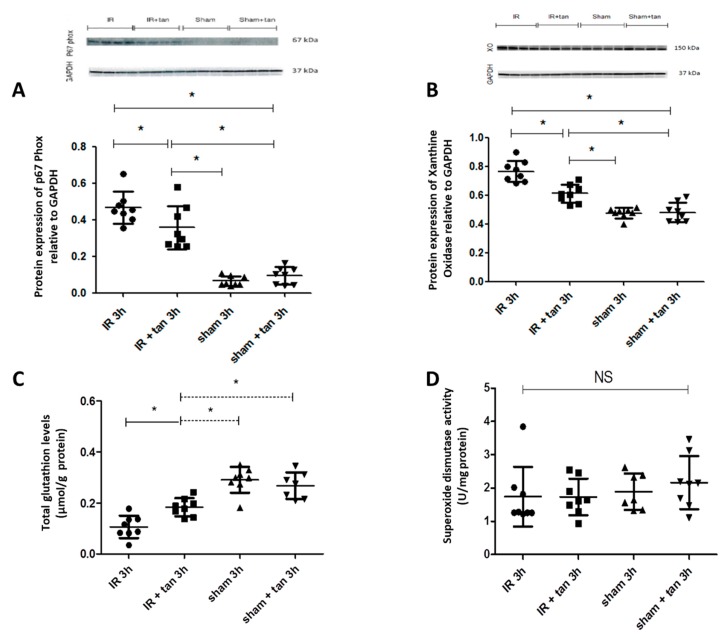
Tannic acid injection protects from oxidative stress development. In kidneys biopsies obtained 3 h after reperfusion following warm ischemia, tannic acid limits P67phox (**A**) and xanthine oxidase (**B**) expressions, and reduces the decrease in total glutathion (**C**) induced by ischemia–reperfusion injury, without effect on superoxide dismutase activity (**D**), *n* = 8, values are mean ± SD, * p < 0.05.

**Figure 6 biomolecules-10-00439-f006:**
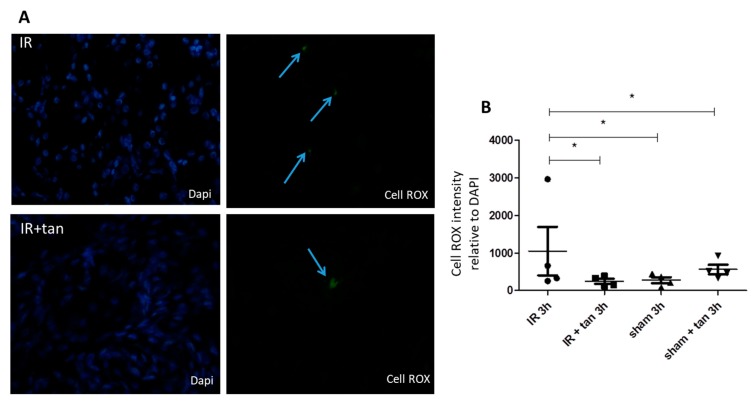
Tannic acid prevents oxidative stress in ischemia–reperfusion kidneys. Reactive oxygen species production induced by ischemia–reperfusion injury was evaluated by cell ROX staining. (**A**) Representative pictures (40X) of staining in kidney biopsies after 3 h of reperfusion following warm ischemia with or without tannic acid injection; (**B**) quantitative evaluation of staining. *n* = 4, values are mean ± SD, * p < 0.05.

**Figure 7 biomolecules-10-00439-f007:**
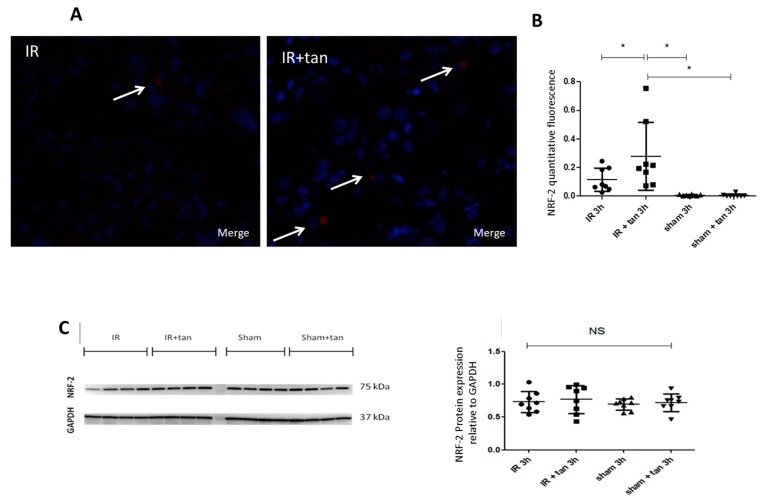
Tannic acid induced Nuclear Factor erythroid-2-Related factor 2 (NRF2) expression in kidneys. Protein expression of NRF2 determined by immunostaining (40X; **A**–**B**) or by western blot (**C**). Quantification of fluorescence intensity for NRF2 immunostaining (**B**). *n* = 8, values are mean ± SD, * p < 0.05.

**Figure 8 biomolecules-10-00439-f008:**
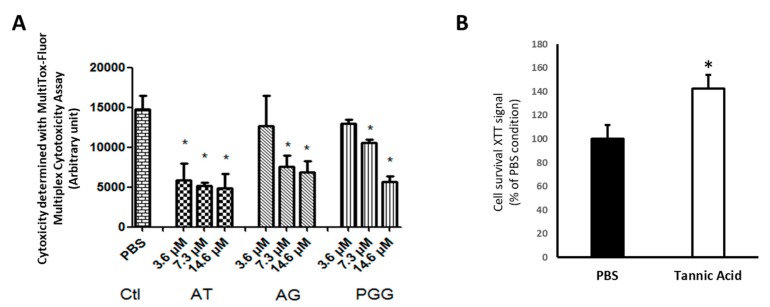
Tannic acid and derivatives protect endothelial cells from hypoxia–reoxygenation injury. (**A**) Cytotoxicity induced by the hypoxia–reoxygenation sequence was prevented by tannic acid (AT) and its main metabolites gallic acid (GA) and penta-O-galloyl-beta-D-glucose (PGG) with a dose-dependent response. Molecules were added in the culture medium during the hypoxia–reoxygenation sequence. *n* = 3, *n* = 2. (**B**) XTT analysis showing that cytotoxicity induced by the hypoxia–reoxygenation sequence was prevented by tannic acid (12.5 mg/mL, 7.3 µmol/L). Molecules were added in the culture medium during the hypoxia–reoxygenation sequence. *n* = 3, *n* = 2, values are mean ± SD, * p < 0.05 to the PBS condition (vehicle).

**Figure 9 biomolecules-10-00439-f009:**
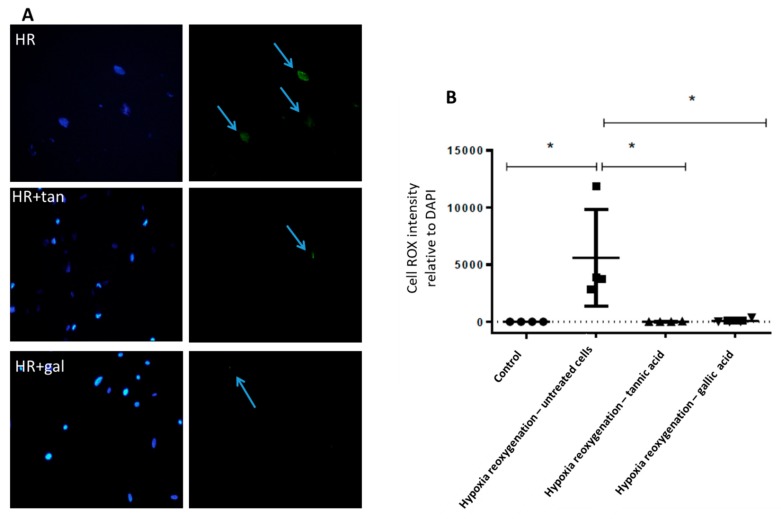
Tannic acid and gallic acid prevent oxidative stress induced by hypoxia–reoxygenation sequence. Reactive oxygen species production was determined by Cell ROX staining. (**A**) Representative pictures (40X) of staining in human aortic endothelial cells (HAEC) subjected to the hypoxia–reoxygenation sequence with or without tannic acid (7.3 µmol/L) or gallic acid (7.3 µmol/L); (**B**) quantitative evaluation of the staining. *n* = 4, values are mean ± SD, * p < 0.05.

**Figure 10 biomolecules-10-00439-f010:**
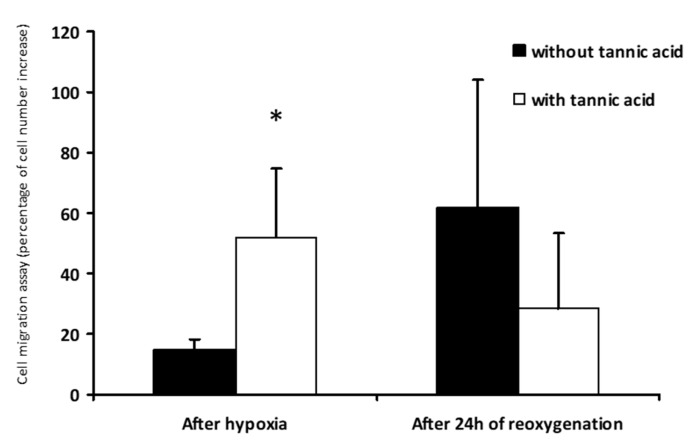
Tannic acid improves endothelial cell migration during hypoxia. Cell migration assay was determined by the number of cells migrating in an area after warm hypoxia, or 24 h of reoxygenation with or without tannic acid (7.3 µmol/L). *n* = 3, values are mean ± SD, * p < 0.05.

**Figure 11 biomolecules-10-00439-f011:**
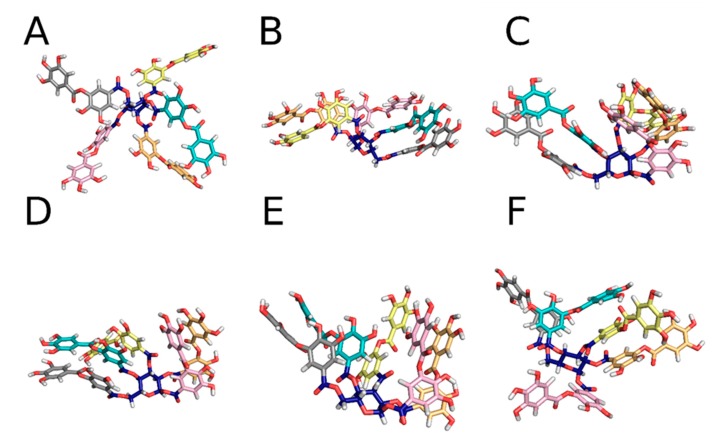
The starting conformation (**A**) and the different representative structures of tannic acid in the solution (**B**–**F**). The PGG is in blue, the five different gallic acid tails are in different colours. Oxygen atoms are in red, and hydrogen atoms in white.

**Figure 12 biomolecules-10-00439-f012:**
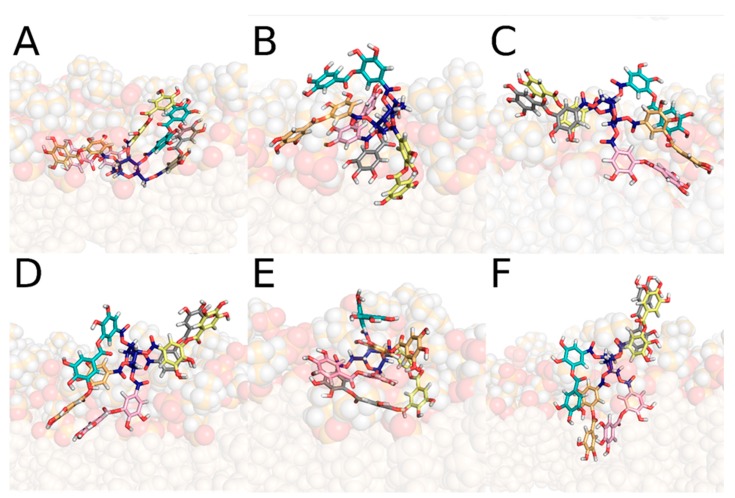
The six different representative structures of tannic acid in the membrane, according to the clustering analysis. The PGG is in blue, the five different gallic acid tails are in different colours. Oxygen atoms are in red, and hydrogen atoms in white. The membrane is depicted as spheres with the phosphatidylcholine headgroups in dark orange with hydrogens and oxygens; the tail carbon atoms are pale orange spheres.

**Figure 13 biomolecules-10-00439-f013:**
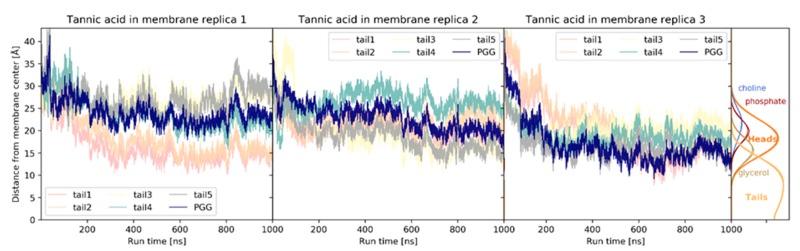
The z-coordinate of the center of mass of different parts of the tannic acid, and positions of the different parts of the membrane.

**Figure 14 biomolecules-10-00439-f014:**
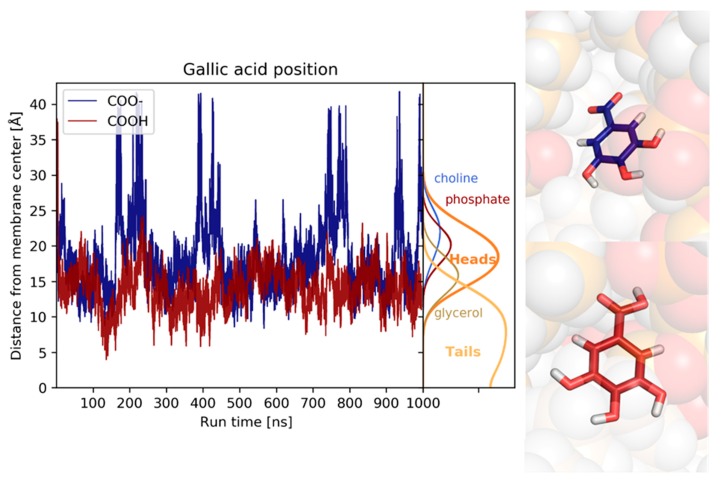
The z-coordinate of the center of mass of the protonated (red) and deprotonated (blue) gallic acid.

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
