# Peer review of "Tannic Acid Improves Renal Function Recovery after Renal Warm Ischemia–Reperfusion in a Rat Model"

_biomolecules, 2020, doi:10.3390/biom10030439_

Round 1
Reviewer 1 Report
The study provides an advance in the field. The paper has a logical flow, starting by measuring blood tannic acid concentration, tannic acid derivates determination, and continuing with the evaluation of tannic acid effects on oxidative stress in vivo, in vitro and in silico, on endothelial cells proliferation. The methods clear and replicable. The results are clearly and appropriately presented.
The conclusions mentioned that tannic acid reduces oxidative stress but not that it stimulated endothelial cells proliferation. This is as important as the antioxidant effect for the recovery stage. I suggest to include this result in the conclusions too.

Author Response
Dear Editor,
Enclosed please find the revision of our manuscript entitled “Tannic acid improves renal function recovery after renal warm ischemia reperfusion in a rat model”. A number of changes have been made to the paper; as indexed in this letter and highlighted in yellow in the manuscript. We wish to thank the reviewers for the constructive and helpful comments. More specific responses are outlined below.
Reviewer 1
1) The conclusions mentioned that tannic acid reduces oxidative stress but not that it stimulated endothelial cells proliferation. This is as important as the antioxidant effect for the recovery stage. I suggest to include this result in the conclusions too.
Thank you for this very relevant point. The conclusion section has been edited. We modified the sentences as: “In addition to the ROS scavenging activity of polyphenols, we suggested, here, a stimulation of endothelial cells proliferation associated to NRF2 process induction.”
Author contributions
This section has been edited. We added : “LA : conception and design, collection and/or assembly of data, data analysis, manuscript writing. FF : conception and design, assembly of data, data analysis, manuscript writing. PC : provision of study material, collection and/or assembly of data, data analysis, manuscript writing. SI: conception and design, collection and/or assembly of data, data analysis, manuscript writing. PAF: modified the general structure of the manuscript to make it more readable. CO : collection and/or assembly of data. RT : conception and design, collection and/or assembly of data, data analysis, manuscript writing. BB : conception and design, data analysis, financial support. PT : conception and design, provision of study material, collection and/or assembly of data, data analysis, manuscript writing. JG : conception and design, provision of study material, manuscript writing, financial support, administrative support. TH : conception and design, data analysis, financial support, administrative support. All authors have read and approved the manuscript.”
Legend Figure
We corrected a mistake in the legend of figure 4. We modified as :”Figure 4: Tannic acid injection was ineffective to improve renal function recovery after cold ischemia- reperfusion in kidney graft model with 6h of cold preservation. Rats were subjected to sham or kidney transplantation with bilateral nephrectomy with or without prior injection of Tannic acid. Renal function was evaluated by creatinine plasma levels concentrations. n=7-8 in transplanted groups and n=6 in sham groups, values are mean ± SD, * p < 0,05.”
Respectfully submitted Jérome Guillard
Reviewer 2 Report
- The first paragraph of Introduction (lines 41-47) should be removed from the body text as they do not serve any purpose, not they are based on references other than the authors’ opinions.
- Legends of Figure 1-14 must be rewritten, as they are part of the original image. Hence losing in dpi.
- Lines 69-70. Remove that sentence, it is redundant. In other words, polyphenols including tannic acid or its derivatives could stimulate 69 the antioxidant defenses in addition to their direct ROS scavenging capacities
- Line 146. Change superoxide to superoxide. In addition, there are more than three instances in the text that the abbreviations are repeated.
- Is there an explanation for such high variation in creatinine levels on Day 4 after cold IRI?
- figure 7 needs to be redesigned in higher dpi, everything is too blurry.
- All ROS staining results are not convincing. Since the authors use the Green Cell ROX system, I want to see fluorescence levels obtained by a fluorescence reader of at least the aortic cells experiment (Figure 9) with and w/o tannic, or gallic acid.
Author Response
Dear editor,
Enclosed please find the revision of our manuscript entitled “Tannic acid improves renal function recovery after renal warm ischemia reperfusion in a rat model”. A number of changes have been made to the paper; as indexed in this letter and highlighted in yellow in the manuscript. We wish to thank the reviewers for the constructive and helpful comments. More specific responses are outlined below.
Reviewer 2
1) The first paragraph of Introduction (lines 41-47) should be removed from the body text as they do not serve any purpose, not they are based on references other than the authors’ opinions.
Indeed, this paragraph needs to be removed because it is useless. We withdraw these sentences.
2) Legends of Figure 1-14 must be rewritten, as they are part of the original image. Hence losing in dpi.
The legends of Figures was rewritten and separated to the different graphs to improve quality of Figures. See attached document
3) Lines 69-70. Remove that sentence, it is redundant. In other words, polyphenols including tannic acid or its derivatives could stimulate the antioxidant defenses in addition to their direct ROS scavenging capacities
This is a relevant suggestion, we removed this sentence.
4) Line 146. Change superoxyde to superoxide. In addition, there are more than three instances in the text that the abbreviations are repeated.
Thank you for pointing out this mistake. We corrected it and used abbreviation as need.
5) Is there an explanation for such high variation in creatinine levels on Day 3 after cold IRI?
The kidney transplantations in rodent were performed by an experiment surgeon (urologist). However, this remains a difficult microsurgery. Day 3 after transplantation is fully into the graft recovery period, and while there are all identical, some rats have slightly faster recoveries period than others, explaining the high variation in creatinine levels on day 3.
6) figure 7 needs to be redesigned in higher dpi, everything is too blurry.
The figure 7 have been redesigned to improve quality in dpi. See attached document
7) All ROS staining results are not convincing. Since the authors use the Green Cell ROX system, I want to see fluorescence levels obtained by a fluorescence reader of at least the aortic cells experiment (Figure 9) with and w/o tannic, or gallic acid.
We respectfully disagree with the reviewer. This project used the Cell ROX system already used to investigate the effect of antioxidative drugs, in a great number of studies in the literature and also by our lab (Soussi et al. Int J Mol Sci. 2019;20(9); Caillaud et al. Neuropharmacology. 2018;139:98-116; Melis et al. J Am Soc Nephrol.2017 Mar;28(3):811-822; Harkcom et al. Proc Natl Acad Sci U S A. 2014;111(24):E2443-52; Mordwinkin et al. Endocrinology. 2012;153(5):2189-97). Moreover to clarify this point, we added in the materials and methods section, a precision about the method used to evaluate the absorbance of Cell ROX system: “The fluorescence of Cell ROX assay was determined by image J software on 7 fields obtained from an Olympus BX41 microscope and reported to the number of cells evaluated by DAPI fluorescence”. As expected, we provide in the table below, the basal levels of fluorescence in the 4 groups found in the Figure 9, with or without tannic acid, or gallic acid. These data show a significant difference of the fluorescence between hypoxia/reoxygenation untreated cells and both treatments as already found with levels relative to DAPI. Hypoxia reoxygenation sequence enhances mortality promoting a high ratio in H/R untreated cells group presented in the figure 9.
Control |
Hypoxia reoxygenation untreated cells |
Hypoxia reoxygenation tannic acid |
Hypoxia reoxygenation gallic acid |
9937 |
18111 |
11244 |
10503 |
9937 |
35093 |
9988 |
10127 |
9941 |
46832 |
10050 |
12589 |
9943 |
21281 |
10260 |
9960 |
Author contributions
This section has been edited. We added : “LA : conception and design, collection and/or assembly of data, data analysis, manuscript writing. FF : conception and design, assembly of data, data analysis, manuscript writing. PC : provision of study material, collection and/or assembly of data, data analysis, manuscript writing. SI: conception and design, collection and/or assembly of data, data analysis, manuscript writing. PAF: modified the general structure of the manuscript to make it more readable. CO : collection and/or assembly of data. RT : conception and design, collection and/or assembly of data, data analysis, manuscript writing. BB : conception and design, data analysis, financial support. PT : conception and design, provision of study material, collection and/or assembly of data, data analysis, manuscript writing. JG : conception and design, provision of study material, manuscript writing, financial support, administrative support. TH : conception and design, data analysis, financial support, administrative support. All authors have read and approved the manuscript.”
Legend Figure
We corrected a mistake in the legend of figure 4. We modified as :”Figure 4: Tannic acid injection was ineffective to improve renal function recovery after cold ischemia- reperfusion in kidney graft model with 6h of cold preservation. Rats were subjected to sham or kidney transplantation with bilateral nephrectomy with or without prior injection of Tannic acid. Renal function was evaluated by creatinine plasma levels concentrations. n=7-8 in transplanted groups and n=6 in sham groups, values are mean ± SD, * p < 0,05.”
Respectfully submitted Jérome Guillard
Round 2
Reviewer 2 Report
I am satisfied with the authors' corrections.